# β-hydroxy-β-methylbutyrate-enriched nutritional supplements for obese adults during weight loss: study protocol of a randomised, double-blind, placebo-controlled clinical trial

Xiaofan Jing,[1] Yuxiang Liang,[2] Renjie Wang,[2] Hongbo Fu,[3] Jiaojiao Jiang,[2,4] Ming Yang ![ORCID] [3,5]

For numbered affiliations see end of article.

**Correspondence to**
Jiaojiao Jiang;
jiangjiaojiao1997@163.com and
Dr Ming Yang;
yangmier@gmail.com

## ABSTRACT

**Introduction** Obesity is a public health challenge worldwide. To maintain a healthy weight, dieting and lifestyle changes are the first-line interventions; however, these interventions are of poor compliance and may cause body composition changes, mainly skeletal muscle wasting (sarcopenia). Dietary supplements for improving body composition while inducing weight loss are therefore needed. β-hydroxy-β-methylbutyrate (HMB) has been proven to be effective for improving muscle mass and muscle strength in athletes, older adults and patients with cancer. We aim to evaluate the effectiveness and safety of HMB-enriched nutritional supplements for improving muscle mass and muscle function in obese adults during calorie restriction.

**Method and analysis** A total of 72 Chinese adults with obesity will be randomised to receive HMB-enriched nutritional supplements (65 g/day) or a placebo for 12 weeks. Participants in both groups will also receive calorie restrictions based on the individualised nutrition guidance of dietitians. Participants and investigators will be blinded to the allocations. The primary outcome will be the mean change in whole-body skeletal muscle mass (measured by bioelectrical impedance analysis). The secondary outcomes will include the mean change of appendicular skeletal muscle mass, body fat mass, basal metabolic rate, phase angle, muscle function and serum biomarkers. The enrolment will commence in December 2021 and will proceed until March 2022.

**Ethics and dissemination** This protocol has been approved by the Biomedical Ethics Committee of West China Hospital (2021-771). All potential subjects will be required to sign a written informed consent. The results of this study will be reported in peer-reviewed academic journals and conferences.

**Trial registration number** NCT04953936.

## INTRODUCTION

Obesity is a chronic disease featured by abnormal or excess body fat that impairs health.[1] Over the past three decades, obesity has become a common health problem that plagues modern people worldwide.[2] For example, more than one-third (36.5%) of the US adult population is obese defined by a body mass index (BMI) $\geq 30 \, \text{kg/m}^2$.[3] In China, the prevalence of abdominal obesity (defined by a waist circumference $\geq 90$ and 85 cm for men and women, respectively) among adults is approximately 29%, and the number of adults with abdominal obesity (using the same definition) is approximately 277.8 million.[4] Obesity is a risk factor for cardiovascular diseases, diabetes, dyslipidaemia, chronic kidney disease and some types of cancer, and increases the risk of psychological disorders, disability, poor quality of life and death.[1,2,5,6] Therefore, obesity prevention and control have become one of the key contemporary public health challenges.[7]

It is well known that increasing physical activity and improving dietary strategies are the keys to the prevention and management of obesity.[1] Many dietary strategies have been proposed for weight loss, such as very low energy diets, ketogenic diets, high-protein diets and high-fibre diets. However, individuals who apply these approaches

may experience weight loss accompanied by body composition changes, mainly skeletal muscle wasting (sarcopenia).[3] Sarcopenia may result in a lower resting metabolic rate and a greater tendency to regain body fat. Recently, there is growing evidence that certain nutritional fortifications can improve body composition during weight loss.[8] A variety of over-the-counter dietary supplements for this purpose are available, and they have gained popularity among consumers for weight management.[9]

Leucine, a branched-chain essential amino acid, and its active metabolite β-hydroxy-β-methylbutyrate (HMB) play an important role in regulating protein synthesis in muscle cells, especially through activation of the mammalian target of rapamycin signal pathway.[10] HMB can also suppress protein degradation pathways by inhibiting intracellular inflammation and caspase-8 activation.[11] However, the rate of conversion of leucine to HMB is very low, and only approximately 5% of leucine in humans is converted to HMB.[12] Studies have found that HMB has the effect of increasing muscle mass and reducing exercise injury.[13 14] The tolerability and safety of HMB in humans have been well established.[15–17] As early as 1995, the US Food and Drug Administration has listed HMB in 'Generally Recognized as Safe' ingredients. In 2010, China's former Ministry of Health announced the approval of HMB as a 'new resource food', which refers to a newly developed, newly discovered or newly introduced food that is not traditionally consumed in China and that meets the basic requirements for food while being non-toxic and harmless to humans.

Studies have shown that HMB supplementation is also beneficial in building muscle strength and may be beneficial in improving body composition. For example, a systematic review that included seven randomised controlled trials (RCTs) showed that HMB supplementation was valuable for maintaining muscle mass in older adults and preventing muscle atrophy due to bed rest or other factors.[18] Another systematic evaluation that included nine RCTs also showed that HMB supplementation improved body composition in older adults.[19] Daily HMB supplementation is supposed to increase the benefits of exercise and improve body composition in healthy adults in the early stages of regular exercise.[20] Therefore, HMB is expected to be a muscle-protective nutritional fortification that can effectively improve muscle mass and muscle strength in both healthy adults and the elderly. However, evidence regarding the effectiveness of HMB on maintaining skeletal muscle in obese adults during weight loss is limited in the literature. Additionally, very few clinical trials regarding HMB have been conducted in Chinese populations. Therefore, this trial aims to assess the effectiveness and safety of HMB-enriched nutritional supplements for improving muscle mass and muscle function in Chinese adults with obesity during the weight loss process using calorie restriction.

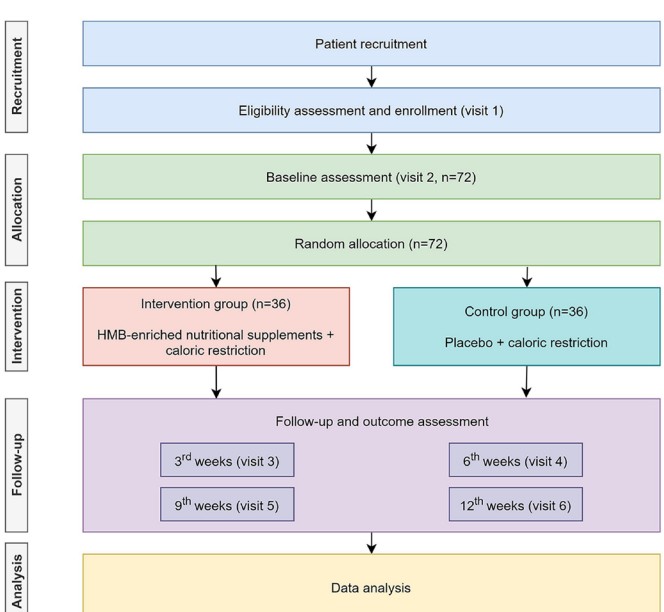

**Figure 1** Flow chart of this trial. HMB, β-hydroxy-β-methylbutyrate.

## METHODS AND ANALYSIS

### Study design

We designed a single-centre, randomised, parallel-group, double-blind, placebo-controlled trial. We plan to recruit 72 Chinese obese adults who intend to perform weight loss via calorie restriction. The participants will be randomly assigned to either the intervention group or the control group at a 1:1 ratio and will receive HMB-enriched nutritional supplements or placebo, respectively, for 12 weeks. The flow chart of this trial is presented in figure 1 while the overall schedule of this trial is shown in table 1.

### Recruitment

This trial will be conducted at West China Hospital, Sichuan University, China. To reach the target sample size, we plan to advertise this trial using the hospital's homepage, WeChat and flyers inside and outside the hospitals.

### Participants

#### Inclusion criteria

Individuals who meet all the following criteria will be invited to participate in this study: (1) men and women aged 30–50 years; (2) currently having obesity defined by BMI $\geq 28 \, \text{kg/m}^2$ (based on measured body weight (BW) and height at booking using validated scales)[21]; (3) having an intention to lose weight via calorie restriction; (4) healthy and able to walk independently; (5) able to eat independently to meet their energy needs; (6) without significant BW change in the last 6 months (less than 5% change in BW); (7) having sedentary habit defined by Sedentary Behavior Research Network[22]; (8) able to collaborate with the research staff. Individuals will be recruited by research nurses or physicians. The study protocol will be provided in advance and individuals who wish to participate will be offered the opportunity

**Table 1** Schedule of enrolment, interventions and assessments

| Timepoint | Enrolment | Allocation | Postallocation | | | |
|---|---|---|---|---|---|---|
| | Visit 1 | Visit 2 | Visit 3 | Visit 4 | Visit 5 | Visit 6 |
| | Screening (−4 to 0 week) | Baseline (0 week) | 3 weeks±2 days | 6 weeks±2 days | 9 weeks±2 days | 12 weeks±2 days |
| **Enrolment** | | | | | | |
| Consent | √ | | | | | |
| Inclusion and exclusion | √ | | | | | |
| Demographics | √ | | | | | |
| Medical history | √ | | | | | |
| Body height | √ | | | | | |
| Body weight | √ | | | | | |
| Blood pressure | √ | | | | | |
| Heart rate | √ | | | | | |
| BIA | √ | | | | | |
| Allocation | | √ | | | | |
| **Interventions*** | | | | | | |
| HMB nutritional supplement+calorie restriction | | ●————————————————————————● | | | | |
| Placebo+calorie restriction | | ●————————————————————————● | | | | |
| **Assessments** | | | | | | |
| Inclusion and exclusion | | √ | | | | |
| Body weight | | √ | | √ | | √ |
| Blood pressure | | √ | | √ | | √ |
| Heart rate | | √ | | √ | | √ |
| WC | | √ | | √ | | √ |
| HC | | √ | | √ | | √ |
| IPAQ-SF | | √ | √ | √ | √ | √ |
| FFQ | | √ | √ | √ | √ | √ |
| BIA | | √ | | √ | | √ |
| Sit-to-stand test | | √ | | √ | | √ |
| HGS | | √ | | √ | | √ |
| Blood tests† | | √ | | | | √ |
| Adverse events‡ | | | √ | √ | √ | √ |
| Adherence‡ | | | √ | √ | √ | √ |

*Both intervention groups will also receive dietary instruction.
†The blood tests will include glucose metabolism (fasting glucose and insulin), albumin, haemoglobin, alanine aminotransferase, aspartate aminotransferase, gamma-glutamyl transpeptidase, cystatin C, creatinine, estimated glomerular filtration rate, lipid profiles and full blood counts.
‡Weekly WeChat or telephone contacts will be performed to assess adverse events and adherence (including dietary compliance).
BIA, bioimpedance analysis; FFQ, Food Frequency Questionnaire; HC, hip circumference; HGS, hand grip strength; HMB, β-hydroxy-β-methylbutyrate; IPAQ-SF, International Physical Activity Questionnaire Short Form; WC, waist circumference.

to discuss the study and consent to participate. All participants will be required to sign the written informed consent before participation.

### Exclusion criteria

Individuals with any of the following criteria will be excluded: (1) a history of intolerance to enteral nutrition, food (eg, lactose intolerance) or being allergic to the components of the HMB-enriched nutritional supplements (eg, soy or corn); (2) having any implants (eg, pacemakers, implantable cardioverter defibrillators and dental implants); (3) current use of other nutritional supplements or drugs that may have an impact on the effectiveness of the intervention (eg, glucocorticoids, antineoplastic drugs, antituberculosis drugs, sedatives, antipsychotics, muscle relaxants, monoamine oxidase inhibitors, antibiotics) within 3 months prior to enrolment; (4) secondary obesity

 

caused by diseases (eg, hypothalamic diseases and hypercortisolism) or drugs (eg, glucocorticoids, insulin, tricyclic antidepressants and weight loss supplements/drugs); (5) clinically visible oedema; (6) trauma, surgery, hospitalisation, fall or fracture within 6 months prior to enrolment; (7) pregnancy, having pregnancy plans or lactation; (8) individuals who are participating in other clinical trials; (9) any acute illness (eg, acute infection, myocardial infarction, arrhythmia, myocarditis, appendicitis, etc); (10) swallowing disorders; (11) abnormal thyroid function (including hyperthyroidism and hypothyroidism); (12) history of diabetes, respiratory diseases, cardiovascular diseases, uncontrolled hypertension, kidney diseases, digestive system diseases, renal insufficiency, mental illness, neurological diseases, haematologic diseases, liver diseases (except for fatty liver), chronic infection (eg, tuberculosis), immune system diseases, joint disorders, any type of tumour; (13) alcohol consumption over two standard alcoholic beverages (20 g of alcohol/day); (14) menopause women; (15) difficulty to comply with the study protocol; and (16) other conditions that indicate the individuals are inappropriate for participation in this study.

### Withdrawal criteria

We will withdraw participants from this trial after randomisation for the following reasons: (1) if the participants request to withdraw from this trial; (2) if there is evidence of a complication or the occurrence of an unknown disease, the investigators can decide that the patient cannot continue in the study; (3) if there is an adverse event (AE) classified as grade 3 or higher according to the Common Terminology Criteria for Adverse Events (CTCAE) version 5.0; (4) if investigators believe the participants are unsuitable for continuing participating in the study for any other reason.

### Randomisation and allocation concealment

We will use the stratified block randomisation method to perform randomisation. A random number sequence will be automatically generated by R V.4.0.3 (R Foundation for Statistical Computing, Vienna, Austria) before the inclusion of the first participants. Gender is first used as a stratification factor, and 36 cases of each gender will be included, and then block group randomisation will be performed for each gender, with block group lengths set to 2–8 and randomly selected by the computer.

To perform allocation concealment, we will seal the randomisation list in sequentially numbered opaque envelopes. The envelopes will be opened sequentially at the beginning of the intervention in the order of entry, and the grouping of participants will be determined according to the allocation scheme inside the envelope.

### Blinding

By using identical doses and appearance of HMB-enriched nutritional supplements and placebos, participants and outcome assessors will be blinded to group allocation.

### Interventions

The participants will be randomly assigned into two groups: the intervention group and the control group. The two groups will be randomly labelled A or B. The participants in both groups will perform a calorie restriction diet. They will receive individualised nutritional guidance from professional dietitians by video regarding calorie restriction strategies. The principle for calorie restriction will be based on the equation: energy intake=ideal BW (ie, height (cm)−105)×25 kcal/kg/day.[23] The compliance will be followed via a WeChat group and telephone. The intervention duration in both groups will be 12 weeks.

### Intervention group

The participants in the intervention group will receive oral HMB-enriched nutritional supplements (65 g, once daily) in powder form packaging in opaque plastic bottles, which include soybean isolate, protein, whey protein, casein, HMB, L-glutamine, wheat oligopeptide, medium-chain fatty acids, vitamin D, vitamin E, konjac powder and fish oil powder.

### Control group

The participants in the control group will receive a placebo (maltodextrin 65 g once daily) in powder form packaging in the same opaque plastic bottles as the intervention.

### Outcomes

As shown in table 1, the assessors, who will be blinded to the group allocation, will ask the participants to complete the questionnaires (including demographic characteristics, physical activity and energy intake), anthropometric measurements (including BW, height, waist circumference and hip circumference), heart rate, blood pressure, hand grip strength (HGS) measurement, sit-to-stand test and body composition at the baseline, the 6-week intervention and the end of the 12-week intervention. Additionally, participants will complete the self-reported questionnaires at the 3 and 9 weeks.

Moreover, fasting blood will be taken by a nurse at baseline and at the end of the 12-week intervention after intervention for laboratory analysis of biomarkers of glucose metabolism (fasting glucose and insulin), albumin, haemoglobin, alanine aminotransferase (ALT), aspartate aminotransferase (AST), gamma-glutamyl transpeptidase (GGT), cystatin C, creatinine, estimated glomerular filtration rate (eGFR), lipid profiles and full blood counts. The fasting blood will be sent to the clinical laboratory of West China Hospital, Sichuan University, within 2 hours after withdrawal. These biomarkers will be tested using standard methods. Moreover, the homeostasis model assessment of insulin resistance (HOMA-IR) will be calculated using the equation: fasting glucose (mg/dL)×fasting insulin (μIU/mL)/405.

### Primary outcome

The mean change in whole-body skeletal muscle mass (SMM) from the beginning to the end of the 12-week

intervention will be measured for all participants. The whole-body lean body mass will be measured by trained research staff using a bioimpedance analysis device (BIA; InBody 770; Biospace, Seoul, Korea). The participants will be guided through the BIA test. They will be asked to stand upright at least 5 min before testing, then to step on the footplate barefoot with their heels on the rear sole electrodes and their hand holding the hand electrode. Their arms should be straight and should not touch the side of the body during the test.

## Secondary outcomes
We will measure the following secondary outcomes:
► The mean change in appendicular skeletal muscle mass, trunk skeletal muscle mass, total body fat mass (BFM), appendicular body fat mass, trunk body fat mass, basal metabolic rate, phase angle and BW from the beginning to weeks 6 and 12, as well as the mean change in whole-body SMM from the beginning to week 6, will be measured for all participants. These data will be measured using InBody 770 (Biospace).
► The mean change in HGS from the beginning to weeks 6 and 12 will be measured for all participants. A trained nurse will measure HGS using a digital hand-held dynamometer (EH101; Xiangshan, Guangdong, China) to the nearest 0.1 kg according to the recommendation of the Chinese National Physical Fitness Evaluation Standard.[24] The participants will be asked to perform three measurements for each hand, with rest periods of ≥30 s between each measurement. The HGS will be determined as the maximum value of the six records.
► The mean change in waist circumference and hip circumference from the beginning to weeks 6 and 12 will be measured for all participants. A trained nurse will measure the waist circumference using a tape at the level mid-way between the lower rib margin and the iliac crest with the participant breathing out gently. Hip circumference will be measured as the maximum circumference over the buttocks.[25]
► The mean change in physical activity from the beginning to weeks 3, 6, 9 and 12 will be measured for all participants using the self-reported International Physical Activity Questionnaire Short Form (IPAQ-SF).[26]
► The mean change in the score of the sit-to-stand test from the beginning to weeks 6 and 12 will be measured for all participants. A trained nurse will perform the 30 s sit-to-stand test,[27] which is a classic test for measuring the performance of lower extremity muscles.
► The mean change in energy intake from the beginning to weeks 3, 6, 9 and 12 will be measured for all participants using the self-reported Food Frequency Questionnaire (FFQ).[28]
► The mean change of lipid profiles, fasting glucose, fasting insulin, HOMA-IR and albumin from the beginning to the end of the 12-week intervention will be measured for all participants.

## Safety outcomes
We will record all AEs during the entire study period. The participants will be asked to complete a self-reported questionnaire regarding any AE every 3 weeks. The participants will also be encouraged to report AEs to the research staff via telephone when there is any AE occurrence. We will rank the severity of AEs according to the CTCAE version 5.0. The cause of the AEs will be ranked as unknown, definitely not related, probably not related, possibly related, probably related and definitely related.[29]

Moreover, we will report the mean change in the laboratory tests regarding AEs from the beginning to the end of the 12-week intervention: haemoglobin, AST, ALT, GGT, cystatin C, creatinine and eGFR.

## Feasibility outcomes
We will record the recruitment rate and completion rate to assess the feasibility of this trial. Additionally, the adherence rate to the interventions will be calculated. Moreover, we will record the reasons for participants' ineligibility and for dropping out of this trial.

## Data collection, management and monitoring
Research coordinators will formally obtain the participants' informed consent before collecting and recording data in the case report form (CRF). The assessment of outcomes will be done by assessors who are unaware of the assignment of the groups.

To ensure that participants adhere to the research strategy, all participants will be included in the WeChat group and the investigators will call them at least 1 day beforehand for each appointment and follow-up session. Participants who do not attend follow-up sessions will be asked for their reasons and telephone contacts will be made to encourage adherence to the intervention and attendance at appointments. We will also contact participants via WeChat or telephone once a week to determine if any AEs have occurred and to assess their compliance by asking each participant to complete a daily compliance record and the IPAQ-SF and FFQ questionnaires.

For the study's quality, monitoring and auditing of the data will be carried out. Prior to the beginning of the study, there will be an introductory meeting. At every important point in the trial, such as participant enrolment, the mid-way point and the conclusion, monitoring staff will visit the institution. In addition, monitoring staff will ensure consistency in the documentation of data contained in the CRF and source document, as well as ensuring that the entire study process is conducted according to the protocol.

## Sample size estimation
To date, no studies have been done to determine whether HMB-enriched nutritional supplements are effective and safe for improving muscle mass and muscle function in obese adults during weight loss. To determine the sample size, we examined similar studies on HMB and referred to the study by Peng and Cheng that evaluated

the effectiveness of HMB for improving SMM and quality in prefrail older adults.[12] Using a 0.05 significance level, 80% statistical power and 25% attrition rate, a sample size of 70 has been assumed using an assumed effective size of 0.93. Moreover, to achieve a gender-balanced distribution within and between groups, a random stratification method including 36 samples per gender is proposed, adding up to 72 samples in total.

### Statistical analysis plan

The statistical analysis will be performed by an independent statistical expert blinded to the group allocation. SPSS V.26.0 (IBM) and R V.4.0.3 (R Foundation for Statistical Computing, Vienna, Austria) will be used for statistical analysis.

Each group will be summarised in terms of demographic characteristics and baseline variables. For comparisons of continuous variables, two-sample t-tests or Mann-Whitney U tests will be used, whereas for categorical variables, the $\chi^2$ test or Fisher's exact test will be used.

As aforementioned, the primary outcome will be in SMM from the beginning to the end of the 12-week intervention. The independent two-sample t-test or Mann-Whitney U test will be used based on the distributional normality. We will make adjustments based on the covariates if there are significant demographic differences between the groups or if baseline variables differ significantly. In the primary analysis, the full analysis set (FAS) will be the main data set. Participants initially assigned to either group will be included in the FAS. However, our analysis will not use the data of ineligible participants (ie, those who are wrongly enrolled because of measurement or human error), subjects who received less than 75% of the planned intervention or subjects without the primary outcome measurement. For sensitivity analysis, we will use a protocol-specific set to compare with the results from the FAS. A per-protocol analysis will include participants who achieved the completion of the study without major deviations from protocol, who provided the primary outcome and who received at least 75% of the planned intervention. The rate of protocol compliance in each group will be calculated. A significance level of two-sided p<0.05 will be used. If needed, the multiple imputation method will be adopted for missing data.

We will use the same methods for analysing secondary outcomes as we did for analysing the primary outcome. A paired t-test or Wilcoxon signed-rank test will be used to assess the changes in outcome measures within groups before and after the intervention. To validate trend differences per visit, we will use repeated measures analysis of variance with post hoc tests.

Moreover, subgroup analysis will be done to identify whether gender or Asian Working Group on Sarcopenia-defined sarcopenia[30] at baseline would affect the clinical response to HMB-enriched nutritional supplements.

## ETHICS AND DISSEMINATION

This trial has been registered on ClinicalTrials.gov (reference NCT04953936). This trial will be conducted following the Declaration of Helsinki.[31] This study protocol was drafted according to Standard Protocol Items: Recommendations for Interventional Trials statement.[32] It has been approved by the Biomedical Ethics Committee of West China Hospital (reference 2021-771). Any protocol amendments will be reapproved by the Biomedical Ethics Committee of West China Hospital and will be reflected in the participants' informed consent form. We will obtain signed consent forms from all participants regarding the modification.

We intend to publish the results of this trial in appropriate academic peer-reviewed journals and conferences.

### Patient and public involvement

We did not involve patients or the public in the planning and design of this study.

## DISCUSSION

This study will be a randomised, double-blind, parallel-group, placebo-controlled trial aiming to explore the effectiveness and safety of HMB-enriched nutritional supplements for improving muscle mass and muscle strength in obese adults during the weight loss process using calorie restriction. It will determine the feasibility of future large-scale clinical trials on HMB-based interventions for improving body composition (especially muscle health) in obese adults when performing weight loss plans.

The effectiveness of HMB for improving muscle mass and muscle strength has been well studied in athletes, healthy adults, older adults and patients with cancer.[33] However, we found only two clinical trials addressing the effectiveness of HMB on muscle health in obese adults.[34 35] In 2019, an RCT conducted in Iran demonstrated that supplementing with HMB for 6 weeks reduced weight, waist and abdominal circumference, and even increased strength without resistance training, and without affecting lean mass, in sedentary and overweight women.[34] Most recently, Rondanelli et al[35] conducted an RCT in obese or overweight Italian menopausal women. Using an oral nutrition supplement containing arginine, glutamine and HMB for 4 weeks, they observed that visceral adipose tissue and BFM were statistically significantly reduced, whereas fat-free mass (mainly muscle mass) remained the same.

Our trial has some strengths. First, it will be the first to evaluate the effectiveness of HMB-enriched nutritional supplements for improving muscle health in Chinese obese adults. Second, we will use a double-blinded design to minimise bias due to demand characteristics, the placebo effect and measurement bias. Last, we will focus on muscle mass and fat mass and muscle function (ie, HGS and physical performance).

This trial also has some limitations. First, this will be a single-centre trial and the sample size is relatively small. Therefore, the representativeness of our study sample may be limited. Second, participants in this study may be heterogeneous, given the wide age range. This heterogeneity should be considered when interpreting the results. Third, it might not be possible to completely control the dietary and physical activities of all participants. However, we will record the physical activity and energy intakes and will adjust for them as potential confounders. Fourth, the muscle mass and fat mass will be measured with a BIA device rather than a dual-energy X-ray absorptiometry (DXA) scanner. The hydration status of the participants may affect the results of the BIA. However, a recent study showed that DXA and segmental multifrequency BIA (InBody 770, which will be applied in this trial) were equally accurate for estimating muscle mass and fat mass.[36]

## Trial status

This protocol is version 1.2, written on 9 November 2021. The enrolment will commence in December 2021 and will proceed until March 2022.

**Author affiliations**

[1]Department of Clinical Nutrition, West China Hospital, Sichuan University, Chengdu, Sichuan, China

[2]Rehabilitation Medicine Center, West China Hospital, Sichuan University, Chengdu, Sichuan, China

[3]Center of Gerontology and Geriatrics, West China Hospital, Sichuan University, Chengdu, Sichuan, China

[4]Rehabilitation Medicine Key Laboratory of Sichuan Province, Chengdu, Sichuan, China

[5]National Clinical Research Center for Geriatrics, West China Hospital, Sichuan University, Chengdu, Sichuan, China

**Contributors** Conceptualisation: MY. Methodology: MY, JJ, XJ. Validation: MY, JJ. Formal analysis: JJ, XJ. Investigation: XJ, HF, RW, YL. Resources: MY. Data curation: JJ. Writing–original draft preparation: XJ, MY. Writing–review and editing: JJ. Visualisation: XJ. Supervision: MY. Project administration: JJ. Funding acquisition: MY. All authors have read and agreed to the published version of the manuscript.

**Funding** This trial was supported by the National Key R&D Program of China (grant number: 2018YFC2002104) and the K&D Program of the Sichuan Science and Technology Department (grant number: 2020YFS0573).

**Disclaimer** The sponsor had no role in the design of this trial and will not take part in the methods, data collection and data analysis of this trial.

**Competing interests** None declared.

**Patient and public involvement** Patients and/or the public were not involved in the design, or conduct, or reporting, or dissemination plans of this research.

**Patient consent for publication** Obtained.

**Provenance and peer review** Not commissioned; externally peer reviewed.

**ORCID iD**

Ming Yang http://orcid.org/0000-0002-1875-1692

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
