## [Reviewer comments · BMJ Open]

ARTICLE DETAILS

TITLE (PROVISIONAL)	β -hydroxy- β -methylbutyrate-enriched nutritional supplements for obese adults during weight loss: study protocol of a randomized, double-blind, placebo-controlled clinical trial
AUTHORS	Jing, Xiaofan; Liang, Yuxiang; Wang, Renjie; Fu, Hongbo; Jiang, Jiaojiao; Yang, Ming

VERSION 1 – REVIEW

REVIEWER	Kevin Maki Indiana University Bloomington
REVIEW RETURNED	01-Sep-2021

GENERAL COMMENTS	This paper describes a protocol for a clinical trial evaluating β-hydroxy-β-methylbutyrate (HMB)-enriched nutritional supplements compared with placebo for improving muscle mass and muscle function in obese Chinese adults during caloric restriction. The length of the study is 12 weeks, and while somewhat short for a weight-loss trial, since the primary outcome is to evaluate the effects of HMB on body composition during weight loss, it is acceptable. However, if feasible, it may be preferable to extend the length of the study to 16 or 20 weeks. Please also see my specific comments below: Specific comments to Authors: 1. General comment: there is a mixture of present and past tense used in the paper; suggest checking that the appropriate tense is used throughout2. Abstract: suggest adding to the methods that the measurement of whole-body skeletal muscle mass will be done using bioelectrical impedance analysis; because it appears that the article would be published after commencing the trial, suggest changing the description of the dates to past tense3. Page 3, lines 7-29: because the BMI cutpoints used to define obesity in Asian populations are generally different from those of Western populations, suggest that throughout this paragraph that the BMI cutpoints referred to for the obesity prevalence values are defined4. Bottom of page 3-top of page 4: suggest that reference citations are needed to support the sentences describing the rate of conversion of leucine to HMB and the studies that found that HMB increased muscle mass and reduced exercise energy5. Page 4, Line 20: suggest defining a “new resource food”6. Page 5, Lines 49-52: the citation used to support the use of the definition of obesity in this population (≥ 28 kg/m²) is from 2004. Is there a more current definition to support this cutpoint? In a PubMed search, an article in Chinese (English abstract) published in 2019 verified the cutpoints for overweight and obesity recommended by
---

	the Cooperative Meta-analysis Group of China Obesity Task Force (Gao et al. Zhonghua Liu Xing Bing Xue Za Zhi. 2019;40:1533-1540). 7. Bottom of page 6-top of page 7: suggest that rather than defining a significant body weight change of at least 5% as exclusionary, perhaps consider excluding based on a set kg 8. Page 6, Lines 26-37: suggest specifically mentioning exclusion of weight-loss supplements/drugs 9. Page 8: Lines 33-52: suggest not assigning “A” and “B” within the paper to the HMB and placebo groups, respectively, as this would unblind it; suggest instead simply saying that they will be randomly labeled A or B. Also, will the HMB supplement and placebo be administered as tablets, capsules, a sachet to sprinkle on food, or some other form? 10. Pages 9 -10: will whole-body skeletal muscle mass at 6 weeks be evaluated as another secondary outcome? 11. Page 12, Line 42: it says that monitoring staff will visit the institutions (plural), but it was earlier described as a single center study. Please explain, or change to the singular “institution.” 12. Page 13, Lines 17-20: the random stratification method is described as “36 samples per district”, suggest instead of district refer to these as strata (i.e., gender) 13. Page 13, Lines 55-58: suggest defining illegible (ineligible) participants; also suggest being more specific than the general statement of “subjects without any intervention, or subjects without any outcome measurement” 14. Page 14, Line 9: were subjects excluded from the per protocol analysis if they were missing any outcome (e.g., hand strength test, a lab draw), or just the primary outcome? 15. Page 15, Lines 39-55: what were the ethnicities of the studies described? 16. Page 18, Line 22: the authors might consider making their data available through a data repository 17. Table 1: suggest adding the dietary instruction as a row in the table; suggest describing in a footnote the analytes that will be measured in the blood tests; suggest adding mention in the table (possibly in the footnotes) of the weekly WeChat or telephone contacts to assess AEs and the dietary compliance follow-up via WeChat group and telephone; suggest rather than Group A and Group B labels for the Interventions that these be called HMB nutritional supplement + caloric restriction and Placebo + caloric restriction in the table 18. Page 17, Line 31: the hydration status of the participants is listed in the Article Summary as a limitation, but this limitation was not described in the main body of the paper Minor:  1. In the title, change “obesity adults” to “obese adults;” also suggest inserting commas to read as “randomized, double-blind, placebo-controlled clinical trial” 2. Page 4, Line 12: suggest that “good” is not needed in the statement “good tolerability and safety of HMB” 3. Page 5, line 16: “participates” should be “participants”; “and” should be changed to “or” in “either the intervention group and the control group” 4. Page 5, Line 55: “intention to lost weight” should be “intention to lose weight” 5. Page 6, Line 45: the exclusion for any acute illness is missing a number (9) in the list, and the remaining items will then need to be renumbered
--	---

	6. Page 9, Line 14: “participates” should be “participants” 7. Page 9, Line 53: suggest that “stand right” should be “stand upright” 8. Page 11, Line 32: “will also encourage” should be “will also be encouraged”; “advent event” should be “adverse event”
--	---

REVIEWER	Renata Risi University of Rome La Sapienza
REVIEW RETURNED	14-Oct-2021

GENERAL COMMENTS	The protocol proposed by the authors is interesting and well structured. However, some major concerns should be addressed:  -I suggest to give more information in the introduction about the way leucine and its metabolites act in preserving/ameliorating lean mass, also at the molecular level - I strongly recommend the authors to give more details about the dietary interventions they are going to prescribe. In fact, macronutrient distribution, as well as the grade of calorie restriction, are significant determinants for body composition changes, which is the primary end point of this study. -How the compliance to dietary intervention will be evaluated? Maybe the use of questionnaires to evaluate the amount of calorie/macronutrients intake could be helpful - will also post menopause women be included in the study? -I suggest to include the number of expected drop-out in the calculation of sample size
--

VERSION 1 – AUTHOR RESPONSE

Reviewer: 1

Dr. Kevin Maki, Indiana University Bloomington

Comments to the Author:

This paper describes a protocol for a clinical trial evaluating β -hydroxy- β -methylbutyrate (HMB)-enriched nutritional supplements compared with placebo for improving muscle mass and muscle function in obese Chinese adults during caloric restriction. The length of the study is 12 weeks, and while somewhat short for a weight-loss trial, since the primary outcome is to evaluate the effects of HMB on body composition during weight loss, it is acceptable. However, if feasible, it may be preferable to extend the length of the study to 16 or 20 weeks.

Response: Thank you very much for your time and valuable comments. We plan to perform a 12-week intervention based on previous similar RCTs and our budget.

Please also see my specific comments below:

Specific comments to Authors:

1. General comment: there is a mixture of present and past tense used in the paper; suggest checking that the appropriate tense is used throughout

Response: Thank you. We deeply appreciate you time and valuable comments. We revised the present and past tense throughout the manuscript where necessary.

2. Abstract: suggest adding to the methods that the measurement of whole-body skeletal muscle mass will be done using bioelectrical impedance analysis; because it appears that the article would be published after commencing the trial, suggest changing the description of the dates to past tense

Response: Thank you. We added the information that the measurement of whole-body skeletal muscle mass will be performed using BIA. We changed the date of enrollment because of the COVID-19 pandemic, and we plan to commence the trial Dec 2021.

3. Page 3, lines 7-29: because the BMI cutpoints used to define obesity in Asian populations are generally different from those of Western populations, suggest that throughout this paragraph that the BMI cutpoints referred to for the obesity prevalence values are defined

Response: Thank you. We added the cutoff points of BMI or WC throughout the manuscript when obesity prevalence was present.

4. Bottom of page 3-top of page 4: suggest that reference citations are needed to support the sentences describing the rate of conversion of leucine to HMB and the studies that found that HMB increased muscle mass and reduced exercise energy

Response: Thank you. We added the relevant references according to your comments.

5. Page 4, Line 20: suggest defining a “new resource food”

Response: A new resource food is a newly developed, newly discovered, or newly introduced food without eating habits in Chinese that meets the basic requirements for food while being non-toxic and harmless to humans. We added this definition in the manuscript.

6. Page 5, Lines 49-52: the citation used to support the use of the definition of obesity in this population (≥ 28 kg/m²) is from 2004. Is there a more current definition to support this cutpoint? In a PubMed search, an article in Chinese (English abstract) published in 2019 verified the cutpoints for overweight and obesity recommended by the Cooperative Meta-analysis Group of China Obesity Task Force (Gao et al. Zhonghua Liu Xing Bing Xue Za Zhi. 2019;40:1533-1540).

Response: Thank you very much for your kind help. The cutoff (≥ 28 kg/m²) is widely used in clinical practice in China. The reference paper you mentioned also concluded that “With specificity 90% for identification of risk factors, the appropriate obese cut-off points of BMI were around 28.0 kg/m² both in men and women, with the range of 27.0 to 28.9 kg/m²”. We, therefore, did not change the cutoff.

7. Bottom of page 6-top of page 7: suggest that rather than defining a significant body weight change of at least 5% as exclusionary, perhaps consider excluding based on a set kg

Response: Thank you. We revised this point according to your comments.

8. Page 6, Lines 26-37: suggest specifically mentioning exclusion of weight-loss supplements/drugs

Response: We appreciate your valuable comments. We revised this point accordingly.

9. Page 8: Lines 33-52: suggest not assigning “A” and “B” within the paper to the HMB and placebo groups, respectively, as this would unblind it; suggest instead simply saying that they will be randomly labeled A or B. Also, will the HMB supplement and placebo be administered as tablets, capsules, a sachet to sprinkle on food, or some other form?

Response: Thank you very much. We revised these contents accordingly to your comments. The HMB and placebo will be administered in the same opaque plastic bottle.

10. Pages 9 -10: will whole-body skeletal muscle mass at 6 weeks be evaluated as another secondary outcome?

Response: Thank you. Yes, this will be another secondary outcome. We added this point in this section.

11. Page 12, Line 42: it says that monitoring staff will visit the institutions (plural), but it was earlier described as a single center study. Please explain, or change to the singular “institution.”

Response: Thank you. There will be only one center. We corrected this typo.

12. Page 13, Lines 17-20: the random stratification method is described as “36 samples per district”, suggest instead of district refer to these as strata (i.e., gender)

Response: Thank you. We revised this point accordingly.

13. Page 13, Lines 55-58: suggest defining illegible (ineligible) participants; also suggest being

more specific than the general statement of “subjects without any intervention, or subjects without any outcome measurement”

Response: Thank you. It should ineligible participants (those who are wrongly enrolled because of measurement or human error). We added this definition in the sentence. In addition, we revised the statement of “subjects without any intervention, or subjects without any outcome measurement” as follows: “subjects who received less than 75% of the planned intervention, or subjects without the primary outcome measurement”.

14. Page 14, Line 9: were subjects excluded from the per protocol analysis if they were missing any outcome (e.g., hand strength test, a lab draw), or just the primary outcome?

Response: Just the primary outcome. We revised this sentence accordingly.

15. Page 15, Lines 39-55: what were the ethnicities of the studies described?

Response: Thank you for your comments. These studies were conducted in Iran and Italy. We added this information in this paragraph.

16. Page 18, Line 22: the authors might consider making their data available through a data repository

Response: Thank you. Our data will not be uploaded to a public repository. However, we are willing to share our data upon reasonable request.

17. Table 1: suggest adding the dietary instruction as a row in the table; suggest describing in a footnote the analytes that will be measured in the blood tests; suggest adding mention in the table (possibly in the footnotes) of the weekly WeChat or telephone contacts to assess AEs and the dietary compliance follow-up via WeChat group and telephone; suggest rather than Group A and Group B labels for the Interventions that these be called HMB nutritional supplement + caloric restriction and Placebo + caloric restriction in the table

Response: Thank you for your suggestion. 1) We added a footnote in Table 1 to describe the dietary instruction instead of adding the dietary instruction as a row because the latter might mislead the readers that there were three groups. 2) We added a footnote to clarify the details of blood tests. 3) We added a footnote to state that ‘Weekly WeChat or telephone contacts will be performed to assess adverse events and adherence (including dietary compliance).’ 4) We revised the labels of Group A and Group B according to your comments.

18. Page 17, Line 31: the hydration status of the participants is listed in the Article Summary as a limitation, but this limitation was not described in the main body of the paper

Response: Thank you. We added this point in the limitation section.

Minor:

1. In the title, change “obesity adults” to “obese adults;” also suggest inserting commas to read as “randomized, double-blind, placebo-controlled clinical trial”

Response: We appreciate your comments. We revised the title accordingly.

2. Page 4, Line 12: suggest that “good” is not needed in the statement “good tolerability and safety of HMB”

Response: Thank you. We deleted ‘good’ according to your comments.

3. Page 5, line 16: “participates” should be “participants”; “and” should be changed to “or” in “either the intervention group and the control group”

Response: Thank you. We corrected these typos.

4. Page 5, Line 55: “intention to lost weight” should be “intention to lose weight”

Response: Thank you. We corrected this typo.

5. Page 6, Line 45: the exclusion for any acute illness is missing a number (9) in the list, and the remaining items will then need to be renumbered

Response: Thank you very much. We renumbered the list.

6. Page 9, Line 14: “participates” should be “participants”

Response: Thank you. We corrected this typo.

7. Page 9, Line 53: suggest that “stand right” should be “stand upright”

Response: Thank you. We corrected this error.

8. Page 11, Line 32: “will also encourage” should be “will also be encouraged”; “advent event” should be “adverse event”

Response: Thank you very much. We corrected these typos.

Reviewer: 2

Dr. Renata Risi, University of Rome La Sapienza

Comments to the Author:

The protocol proposed by the authors is interesting and well structured. However, some major concerns should be addressed:

-I suggest to give more information in the introduction about the way leucine and its metabolites act in preserving/ameliorating lean mass, also at the molecular level

Response: We appreciate your valuable comments. We added the following information in this section: “Leucine, a branched-chain essential amino acid, and its active metabolite HMB play an important role in regulating protein synthesis in muscle cells, especially through activation of the mammalian target of rapamycin (mTOR) signal pathway ¹⁰. HMB can also suppress protein degradation pathways by inhibiting intracellular inflammation and caspase-8 activation ¹¹”

- I strongly recommend the authors to give more details about the dietary interventions they are going to prescribe. In fact, macronutrient distribution, as well as the grade of calorie restriction, are significant determinants for body composition changes, which is the primary end point of this study.

Response: Thank you very much. We added the details about the dietary interventions according to your comments. “The principle for energy supply ratios of the three major nutrients are as follows: (1) Protein: we recommend an adequate supply of protein (1.2-1.5 g/kg) and using soy protein to partially replace casein; (2) fat: we recommend a 20%-30% energy supply ratio for fat. Also, we recommend increasing foods rich in omega-3 fatty acids; (3) carbohydrate: we recommend determining the supply of carbohydrates based on the intake of protein and fat. The appropriate ratio of carbohydrates for energy supply should be 40%-55%. The source of carbohydrates should be mainly starch-based complex carbohydrates, and the intake of dietary fiber is recommended as 25-30 g/d. Moreover, we recommend strictly limiting the intake of simple sugars (i.e., monosaccharides, disaccharides) in foods or beverages.”

-How the compliance to dietary intervention will be evaluated? Maybe the use of questionnaires to evaluate the amount of calorie/macronutrients intake could be helpful

Response: We appreciate your comments. As we stated in the Interventions section, the compliance will be followed via a WeChat group and telephone. Based on our experience, this method was better than questionnaires.

- will also post-menopausal women be included in the study?

Response: Thank you for your comments. We would exclude menopause women. We added this point in the exclusion criteria.

-I suggest to include the number of expected drop-out in the calculation of sample size

Response: Thank you for your comments. As we stated in the ‘sample size estimation’, we considered 25% attrition rate, which includes expected drop-out.

VERSION 2 – REVIEW

REVIEWER	Kevin Maki Indiana University Bloomington
REVIEW RETURNED	28-Nov-2021

GENERAL COMMENTS	Thank you for your attention to my previous suggestions for revisions. Please see a few additional minor comments below. Specific comments to Authors: 1. In my previous comments I had suggested that the authors be clear about the BMI cut point that was used to define obesity in China. While the Methods section indicates that the cut point for this analysis was 28 kg/m², the cut point for obesity used to determine that approximately 277.8 million in China are obese that is mentioned in the Introduction is not stated. Since the cut point for obesity in the US of 30 kg/m² is mentioned in the prior sentence, this remains unclear.2. In my previous comments I had suggested adding a description of the form of the HMB-enriched supplement and placebo control (e.g., tablets, capsules, powders/sachet, etc.). While the description of the HMB and placebo products was expanded to state that these will be administered in the same opaque plastic bottle, the paper still does not include a description of the form of the supplement.3. In the definition of a “new resource food,” suggest adding an explanation of what is meant by “without eating habits in Chinese”4. In the Exclusion Criteria, suggest that a more detailed specific explanation of “having any implants (e.g., pacemakers)” is needed, since some types of implants may not be exclusionary (e.g., dental implants)5. Suggest that the method of evaluating compliance via the WeChat group and telephone should be described in more detail. In the Statistical Analysis section, it is stated that subjects who receive <75% of the planned intervention will be excluded from the primary analysis, so will there be a formal calculation conducted to assess compliance?6. Minor comment: in the Primary outcome section on page 11, suggest that “5 m” should be change to “5 min”
---

VERSION 2 – AUTHOR RESPONSE

Reviewer: 1

Dr. Kevin Maki, Indiana University Bloomington

Comments to the Author:

Thank you for your attention to my previous suggestions for revisions. Please see a few additional minor comments below.

Specific comments to Authors:

1. In my previous comments I had suggested that the authors be clear about the BMI cut point that was used to define obesity in China. While the Methods section indicates that the cut point for this analysis was 28 kg/m², the cut point for obesity used to determine that approximately 277.8 million in China are obese that is mentioned in the Introduction is not stated. Since the cut point for obesity in the US of 30 kg/m² is mentioned in the prior sentence, this remains unclear.

Response: Thank you for your comments. It should be ‘the number of adults with abdominal obesity (instead of obese) is approximately 277.8 million ‘. Abdominal obesity was defined by the same definition as we mentioned in this sentence. We revised this sentence as follows to make it clearer: “In China, the prevalence of abdominal obesity (defined by a waist circumference \geq 90 cm and 85 cm for men and women, respectively) among adults is approximately 29%, and the number of adults with abdominal obesity (using the same definition) is approximately 277.8 million”.

2. In my previous comments I had suggested adding a description of the form of the HMB-enriched supplement and placebo control (e.g., tablets, capsules, powders/sachet, etc.). While the description of the HMB and placebo products was expanded to state that these will be administered in the same opaque plastic bottle, the paper still does not include a description of the form of the supplement.

Response: Thank you very much. The supplements and placebo are in powder form and packaged in opaque plastic bottles. We added this information in the “Interventions” section.

3. In the definition of a “new resource food,” suggest adding an explanation of what is meant by “without eating habits in Chinese”

Response: Thank you. We rewrote this sentence to make it clearer. In 2010, China’s former Ministry of Health announced the approval of HMB as a “new resource food”, which refers to a newly developed, newly discovered, or newly introduced food that is not traditionally consumed in China and that meets the basic requirements for food while being non-toxic and harmless to humans.

4. In the Exclusion Criteria, suggest that a more detailed specific explanation of “having any implants (e.g., pacemakers)” is needed, since some types of implants may not be exclusionary (e.g., dental implants)

Response: Thank you for your suggestion. It is well-known that pacemakers and implantable cardioverter defibrillators are the contradictions of bioimpedance analysis (BIA). Moreover, any metal implants (including some dental implants) would influence the results of BIA according to the manual of InBody 770 (the BIA device that we will use in this study). Therefore, we will not accept individuals with implants (since many people do not know whether their implants are metal or not). We revised these criteria to make them clearer.

5. Suggest that the method of evaluating compliance via the WeChat group and telephone should be described in more detail. In the Statistical Analysis section, it is stated that subjects who receive <75% of the planned intervention will be excluded from the primary analysis, so will there be a formal calculation conducted to assess compliance?

Response: Thank you for your suggestion. We added the following information: “We will also contact participants via WeChat or telephone once a week to determine if any AEs have occurred and to assess their compliance by asking each participant to complete a daily compliance record and the IPAQ-SF and FFQ questionnaires.” Additionally, in the statistical section, we added the following information: “the rate of protocol compliance in each group will be calculated.”

6. Minor comment: in the Primary outcome section on page 11, suggest that “5 m” should be change to “5 min”

Response: Thank you. We corrected this typo.

VERSION 3 – REVIEW

REVIEWER	Kevin Maki Indiana University Bloomington
REVIEW RETURNED	06-Jun-2022

GENERAL COMMENTS	Thank you for your attention to all of my suggestions for revisions.
--